# DEEP ANTI-REGULARIZED ENSEMBLES

## ABSTRACT

We consider the problem of uncertainty quantification in high dimensional regression and classification, for which deep ensemble have proven to be promising methods. Recent observations have shown that deep ensemble return overconfident estimates outside the training domain, which is a major limitation because shifted distributions are often encountered in real-life scenarios. The principal challenge in solving this problem is to solve the trade-off between increasing the diversity of the ensemble outputs and making accurate in-distribution predictions. In this work, we show that an ensemble of large weights networks fitting the training data are likely to meet these two objectives. We derive a simple and practical approach to produce such ensembles, based on an original anti-regularization term penalizing small weights and a control process of the weight increase which maintains the in-distribution loss under an acceptable threshold. The developed approach does not require any out-of-distribution training data neither any trade-off hyper-parameter calibration. We derive a theoretical framework for the approach and show that the proposed optimization can be seen as a "water-filling" problem. Several experiments in both regression and classification settings highlight that Deep Anti-Regularized Ensembles (DARE) significantly improve uncertainty quantification outside the training domain in comparison to recent deep ensemble and out-of-distribution detection methods. All the conducted experiments are reproducible and the source code is available at `https://github.com/AnonymousAccount3/dare`.

## 1 INTRODUCTION

With the adoption of deep learning models in a variety of real-life applications such as autonomous vehicles (Choi et al., 2019; Feng et al., 2018), or industrial product certification (Mamalet et al., 2021), providing uncertainty quantification for their predictions becomes critical. Indeed, various adaptations of classical uncertainty quantification methods to deep learning predictions have been recently introduced as Bayesian neural networks (Mackay, 1992; Neal, 2012), MC-dropout (Gal & Ghahramani, 2016), quantile regression (Romano et al., 2019) and deep ensembles (Lakshminarayanan et al., 2017; Wen et al., 2020; Wenzel et al., 2020). These methods appear to be quite efficient in predicting the uncertainty in the training domain (the domain defined by the training set), called in-distribution uncertainty (Abdar et al., 2021). However, when dealing with data outside the training distribution, i.e. out-of-distribution data (OOD), the uncertainty estimation often appears to be overconfident (D'Angelo & Henning, 2021; Liu et al., 2021; Ovadia et al., 2019). This is a critical issue, because deep models are often deployed on shifted distributions (de Mathelin et al., 2021; Saenko et al., 2010; Xu et al., 2019); overconfidence on an uncontrolled domain can lead to dramatic consequences in autonomous cars or to poor industrial choices in product design.

The problem to be solved is to increase the output diversity of the ensemble in regions where no data are available during training. This is a very challenging task as neural network outputs are difficult to control outside of the training data regions. In this perspective, contrastive methods make use of real (Pagliardini et al., 2022; Tifrea et al., 2022) or synthetic (Jain et al., 2020; Mehrtens et al., 2022; Segonne et al., 2022) auxiliary OOD data to constrain the network output out-of-distribution. However, these approaches can not guarantee prediction diversity for unseen OOD data as the auxiliary sample may not be representative of real OODs encountered by the deployed ensemble. Another set of methods assumes that the diversity of the ensemble outputs is linked to the diversity of the networks' architectures (Zaidi et al., 2021), hyper-parameters (Wenzel et al., 2020), internal

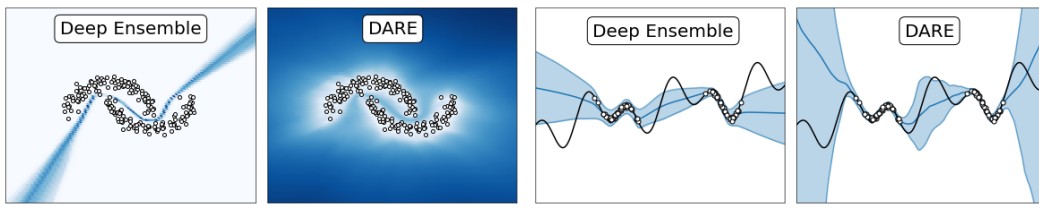

(a) "Two-moons" classification    (b) 1D-regression (Jain et al., 2020)

Figure 1: **Synthetic datasets uncertainty estimation**. White points represent the training data. For each experiment, the ensemble are composed of 20 neural networks. For classification, darker areas correspond to higher predicted uncertainty. For regression, the confidence intervals for $\pm 2\sigma$ are represented in light blue. The full experiment description is reported in Appendix.

representations (Ramé & Cord, 2021; Sinha et al., 2021) or weights (D'Angelo & Fortuin, 2021; Pearce et al., 2018). The main difficulty encountered when using these approaches is to solve the trade-off between increasing the ensemble diversity and returning accurate prediction in-distribution. The current approach to deal with this issue consists in setting a trade-off parameter with hold-out validation (Jain et al., 2020; Liu & Yao, 1999; Pearce et al., 2018) which is time consuming and often penalizes the diversity.

Considering these difficulties, the question that arises is how to ensure important output diversity for any unknown data region while maintaining accurate in-distribution predictions? In this work, we tackle this question with the following reasoning: an ensemble of networks with large weight variance essentially produces large output variance for any data points. Furthermore, to make accurate prediction on the training distribution, the output variance for training data needs to be reduced, which requires that some of the network's weights are also reduced. However, to prevent the output variance from being reduced anywhere other than the training domain, the network weights should then be kept as large as possible. Following this reasoning, we seek an ensemble providing accurate prediction in-distribution and keeping the weights as large as possible.

To meet these two objectives, deviating from traditional recommendations for deep learning training, we propose an "anti-regularization" process that consists in penalizing small weights during training optimization. To find the right trade-off between increasing the weights and returning accurate prediction in-distribution, a control process activates or deactivates the weight increase after each batch update if the training loss is respectively under or above a threshold. Thus, the increase of the weights induces an increase of the prediction variance while the control on the loss enforces accurate in-distribution predictions. Synthetic experiments on toy datasets confirm the efficiency of our proposed approach (cf Figure 1). We observe that the uncertainty estimates of our Deep Anti-Regularized Ensembles (DARE) increase for any data point deviating from the training domain, whereas, for the vanilla deep ensemble, the uncertainty estimates remain low for some OOD regions.

The contributions of the present work are the followings: 1) A novel and simple anti-regularization strategy is proposed to increase deep ensemble diversity. 2) An original control process addresses the trade-off issue between in-distribution accuracy and reliable OOD uncertainty estimates. 3) We provide theoretical arguments to understand DARE as a "water-filling" optimization problem where a bounded global amount of variance is dispatched among the network weights. 4) A new experimental setup for uncertainty quantification with shifted distribution is developed for regression. Experiments are also conducted for out-of-distribution detection on classification datasets.

## 2    DEEP ANTI-REGULARIZED ENSEMBLE

### 2.1    OPTIMIZATION FORMULATION

We consider the supervised learning scenario where $\mathcal{X}$ and $\mathcal{Y}$ are respectively the input and output space. The learner has access to a training sample, $\mathcal{S} = \{(x_1, y_1), ..., (x_n, y_n)\} \in \mathcal{X} \times \mathcal{Y}$ of size $n \in \mathbb{N}$. We consider a set $\mathcal{H}$ of neural networks $h_\theta \in \mathcal{H}$ where $\theta \in \mathbb{R}^d$ refers to the network weights.

We consider a loss function $\ell : \mathcal{Y} \times \mathcal{Y} \to \mathbb{R}_+$ and define the average error of any $h_\theta \in \mathcal{H}$ on $\mathcal{S}$, $\mathcal{L}_\mathcal{S}(h_\theta) = \frac{1}{n} \sum_{(x_i, y_i) \in \mathcal{S}} \ell(h_\theta(x_i), y_i)$.

Traditional deep learning training generally involves the use of weight regularization to avoid overfitting. A penalization term $\mathcal{R}(\theta)$ is added to the average error to form the objective function $\mathcal{L}_\mathcal{S}(h_\theta) + \mathcal{R}(\theta)$ with $\mathcal{R}(\theta)$ increasing with $\theta$ (e.g. $\ell_2$ and $\ell_1$ regularization). However, when used for deep ensemble, such regularization fosters the production of neural networks with small weights, which are then "close" to each others in the weight space and then lack of diversity. The same effect is also induced by the implicit regularization of gradient descent algorithm (Smith et al., 2020). Based on these considerations, we propose the complete opposite approach, which consists in "anti-regularizing" the networks' weights as follows:

$$\min_\theta \ \mathcal{L}_\mathcal{S}(h_\theta) - \lambda \mathcal{R}(\theta). \tag{1}$$

with $\mathcal{R} : \mathbb{R}^d \to \mathbb{R}_+$ a monotone function growing with $\theta$ and $\lambda$ a trade-off parameter. The first term of the optimization objective in Eq. (1): $\mathcal{L}_\mathcal{S}(h_\theta)$ is the loss in-distribution. This term conditions the network to fit the training data which implies smaller in-distribution prediction variances. The second term $-\lambda \mathcal{R}(\theta)$ acts as an "anti-regularization" term which induces an increase of the network weights. This implies a larger variance of the ensemble weights, and therefore a larger prediction variance, especially for data "far" from the training distribution on which the network's predictions are not conditioned. The parameter $\lambda \in \{0, 1\}$ is a binary variable which controls the trade-off between the in-distribution loss and the anti-regularization term. At each batch computation, $\lambda$ is updated as follows:

$$\lambda = \begin{cases} 1 & \text{if } \mathcal{L}_\mathcal{S}(h_\theta) \leq \tau \\ 0 & \text{if } \mathcal{L}_\mathcal{S}(h_\theta) > \tau, \end{cases} \tag{2}$$

with $\tau \in \mathbb{R}$ a predefined threshold.

The underlying idea of the proposed optimization is that, to fulfill both objectives: reducing the loss in-distribution and increasing the weights, large weights will appear more likely in front of neurons which are never or weakly activated by the training data. Therefore, if an out-of-distribution data point activates one of these neurons, large values are propagated through the networks, which induces larger prediction variances. We show, in Sections 2.3 and 4.1, that this claim is supported by theoretical analysis and empirical observations.

The control process is necessary to temper the weight increase, because a large increase of the weights induces an unstable network with reduced accuracy on training data. To be sure to fulfill a performance threshold $\tau$, the weight increase is stopped ($\lambda = 0$) until the loss in-distribution comes back under the threshold. Therefore, the resulting ensemble is composed of networks fitting the training data with weights as large as possible.

## 2.2 Implementation

**Parallel optimization**. Each network of the ensemble is trained with batch gradient descent, independently of the others, with the objective of Eq. (1). This approach allows for parallel training of the ensemble. It is theoretically possible that each network ends up reaching the same optimum resulting in no ensemble diversity. However, we empirically observe that this degenerate case never occurs due to the random process of the optimization and aleatoric weights initialization.

**Regularization function**. We propose the following choice of regularization function:

$$\mathcal{R}(\theta) = \frac{1}{d} \sum_{k=1}^{d} \log(\theta_k^2) \tag{3}$$

With $\theta = (\theta_1, ..., \theta_k)$ the network weights. The use of the logarithmic function is motivated by the "water-filling" interpretation of DARE (cf. Section 2.3).

**Control threshold and Model Saving**. The control threshold $\tau$ should be chosen by the learner based on his targeted error level in-distribution. Smaller $\tau$ leads to smaller increase of the weights. For $\tau = -\infty$, DARE is equivalent to a vanilla deep ensemble. An intuitive value of $\tau$ is close to the validation loss of a vanilla network. We propose, in this work, to set $\tau = \mathcal{L}_{\mathcal{S}_{\text{val}}}(h)(1 + \delta)$ with $\delta > 0$ and $h$ a vanilla network from a deep ensemble.

Regarding the model saving across epochs, we propose to save the model when the validation loss is below $\tau$. Indeed, a small penalization of the validation loss should be accepted to enable the weight increase.

## 2.3 Theoretical Analysis for Linear Regression

The purpose of this theoretical analysis section is to provide an understanding of the underlying dynamic of DARE. We focus our analysis on the linear regression case. This setting offers valuable insights on what happen between two layers of a fully-connected neural network.

We consider the regression problem where $X \in \mathbb{R}^{n \times p}$ and $y \in \mathbb{R}^{n \times 1}$ are respectively the input and output data which compose the training set $\mathcal{S}$. We consider the ensemble of linear hypotheses $\mathcal{H} = \{x \to x^T \theta ; \theta \in \mathbb{R}^p\}$. To simplify the calculation without loosing in generality, we assume that there exists $\theta^* \in \mathbb{R}^p$ such that $\mathcal{L}_{\mathcal{S}}(h_{\theta^*}) = 0$. The loss function is the mean squared error such that for any $h_\theta \in \mathcal{H}$ we have $n\mathcal{L}_{\mathcal{S}}(h_\theta) = ||X\theta - y||_2^2$. We denote $s^2 = (s_1^2, ..., s_p^2) \in \mathbb{R}_+^{* \, p}$ the diagonal of the matrix $\frac{1}{n} X^T X$. We now consider an anti-regularized ensemble $\mathcal{H}_\tau$. To characterize this ensemble, we make the following assumptions for any $h_\theta \in \mathcal{H}_\tau$:

$$\theta \sim \Theta_{\sigma^2} ; \ \mathbb{E}[\theta] = \theta^*, \ \mathrm{Cov}(\theta) = \mathrm{diag}(\sigma^2) \tag{4}$$

$$\mathbb{E}[\mathcal{L}_{\mathcal{S}}(h_\theta)] \leq \delta \, \tau \tag{5}$$

Where $\delta > 0$ and $\mathrm{diag}(\sigma^2)$ is the diagonal matrix of values $\sigma^2 \in \mathbb{R}_+^p$ verifying:

$$\sigma^2 = \operatorname*{arg\,max}_{\sigma^2 = (\sigma_1^2, ..., \sigma_p^2)} \sum_{k=1}^p \log\left(\theta_k^{*2} + \sigma_k^2\right) \tag{6}$$

As presented in Assumption 4, the large weights ensemble distribution is described by the random variable $\theta$ centered in $\theta^*$ with variance $\sigma^2$. Assumption 5 implies that $P(\mathcal{L}_{\mathcal{S}}(h_\theta) \geq \tau) \leq \delta$, by Markov inequality, which models the fact that the loss of each member of DARE is maintained above a threshold $\tau$ thanks to the control process on $\lambda$ (cf Section 2.1). Definition 6 approximates the impact of the anti-regularization term $-\mathcal{R}(\theta)$ in the DARE optimization formulation with an upper bound of $\max_\sigma \mathbb{E}[\mathcal{R}(\theta)]$. The weights are increased as much as possible while the loss stays under the threshold.

Our first theoretical result shows that the weight variance of the anti-regularized ensemble is solution of a "water-filling" optimization problem (Boyd et al., 2006), and is proportional to $1/s^2$, i.e. the inverse of the input features variance.

**Theorem 1.** *There exist a constant $C > 0$ such that for any $k \in [|1, p|]$, the variance of the $k^{th}$ weight component is expressed as follows:*

$$\sigma_k^2 = \max\left[\frac{C \, \delta \, \tau}{s_k^2} - \theta_k^{*2}, 0\right] \tag{7}$$

*Sketch of Proof.* A detailed proof is reported in Appendix. The proof consists in first noticing that $\mathbb{E}[\mathcal{L}_{\mathcal{S}}(h_\theta)] = \sum_{k=1}^p s_k^2 \sigma_k^2$. By combining this result with Assumptions 5 and 6, we show that $\sigma^2$ is solution of the above water filling problem:

$$\begin{aligned} \operatorname*{maximize}_{\sigma^2 \in \mathbb{R}_+^p} \quad & \sum_{k=1}^p \log(\sigma_k^2 + \theta_k^{*2}) \\ \text{subject to} \quad & \sum_{k=1}^p s_k^2 \sigma_k^2 \leq \delta \, \tau \end{aligned} \tag{8}$$

As $\log$ is strictly concave, and the constraints form a compact set on $\mathbb{R}^p$, the problem (8) has a unique solution which is given by (7). $\qquad \square$

The "water-filling" interpretation of the DARE optimization (8) is very insightful: $\delta \, \tau$ is the "global variance capacity" that can be dispatched to the network weights. As, it grows as a function of $\tau$, the more the learner accept a large error in-distribution, the more the global variance capacity increases. We see that each weight component has a different "variance cost" equal to $s_k^2$: for high feature

variance $s_k^2$, the increase of the corresponding weight variance $\sigma_k^2$ penalizes more the training loss. Thus, large weights appear more likely in front of low variance features. Notice also that, when $\theta_k^{*2}$ is high, $\frac{C\,\delta\,\tau}{s_k^2} - \theta_k^{*2}$ is more likely to be negative, leading to $\sigma_k = 0$ (cf Eq. (7)). Besides, $\theta_k^{*2}$ is generally higher for higher $s_k^2$ as it corresponds to more informative feature, enhancing the effect $\sigma_k = 0$ for large $s_k^2$.

We see the importance of choosing a strictly concave function like the logarithm (cf Section 2.2), if instead of log, we choose the identity function for instance, then the solution of (8) degenerates to $\sigma^2 = \left(0, ..., 0, \frac{\delta\,\tau}{s_p^2}\right)$ with $s_p^2$ the lowest feature variance. In this case, all the weight variance is assigned to one component, which reduces the likelihood to detect a deviation of a potential OOD input on another low variance feature.

From Theorem 1, we now derive the expression of the DARE prediction variance for any data $x \in \mathbb{R}^p$:

**Corollary 1.1.** *Let $\mathcal{H}_\tau$ be the large weights ensemble defined by Assumptions 4, 5, 6 and $x \in \mathbb{R}^p$ an input data, the variance of prediction for $h_\theta \in \mathcal{H}_\tau$ is:*

$$\underset{\theta \sim \Theta_{\sigma^2}}{\mathrm{Var}} [h_\theta(x)] = \sum_{k=1}^{p} x_k^2 \max\left[\frac{C\,\delta\,\tau}{s_k^2} - \theta_k^{*2}, 0\right] \tag{9}$$

We deduce from Corollary 1.1 that the prediction variance for $x$ is large when the components $x_k^2$ are large for features with low variance ($s_k^2 \ll 1$). Thus, the predicted uncertainty of DARE is correlated with deviations in directions in which the training input data has small variance. Applied to the hidden-layers of deep fully-connected neural networks, Theorem (1) and Corollary (1.1) suggest that the weight variance is larger in front of neurons weakly activated by the training data. In this case, OOD data that activate such neurons propagate large values inside the network, resulting in a large output variance.

## 3   RELATED WORKS

Increasing ensemble diversity has been an enduring paradigm since the early days of the ensemble learning research. At first, diversity was seen as a key feature for improving the generalization ability of the ensembles (Brown et al., 2005; Kuncheva & Whitaker, 2003; Liu & Yao, 1999; Zhang et al., 2008). Then, with the growing interest in uncertainty quantification, the primary objective of ensemble diversity becomes to produce good uncertainty estimates. In this perspective, a first category of methods propose to increase diversity by using diverse architectures or training conditions among an ensemble of deep neural networks (Lakshminarayanan et al., 2017; Wen et al., 2020; Wenzel et al., 2020; Zaidi et al., 2021). The underlying idea is that the diversity of architecture or local minima reached by the different networks induce a diversity of predictions. Another category of methods propose to explicitly impose a diversity constraint in the loss function of the networks. The loss function is then written $\mathcal{L} + \lambda\mathcal{P}$ where $\mathcal{L}$ is the loss for the task (e.g. mean squared error or negative log-likelihood (NLL)), $\mathcal{P}$ is a penalty term which decreases with the diversity of the ensemble and $\lambda$ is the trade-off parameter between the two terms. Three kinds of penalization are distinguished in the literature. The first kind makes use of training data to compute the penalty term. It includes the Negative Correlation method (NegCorr) (Shui et al., 2018; Zhang et al., 2020) which applies the penalization from (Liu & Yao, 1999) to deep ensembles to enforce a negative correlation between the errors of the networks on the training set. Similarly, (Ross et al., 2020) imposes an orthogonality constraint between the gradients of the ensemble members on training data. Penalizing the similarity between hidden representations of the networks has also been proposed by (Ramé & Cord, 2021; Sinha et al., 2021) using adversarial training. The second kind of penalization refers to contrastive methods that enforces diversity on potential OOD instances rather than training data. This avoids the issue of being over-conditioned by the training domain that can be encountered by previous methods. In this category, several methods suppose that an unlabeled sample containing OOD is available, (Pagliardini et al., 2022; Tifrea et al., 2022). Others avoid this restrictive assumption and simulate potential OOD with random uniform data (Jain et al., 2020; Mehrtens et al., 2022) or instances localized around the training data (Segonne et al., 2022). In the last approach, considered by Anchored-Networks (AnchorNet) (Pearce et al., 2018) and Repulsive Deep Ensemble (RDE) (D'Angelo & Fortuin, 2021), the penalization $\mathcal{P}$ is a function of the network's parameters which

forces the networks to reach local minima spaced from each other in parameters space. Our proposed DARE approach relates particularly to these two methods. Our assumption is that imposing weights diversity has more chance to obtain a global output diversity rather than imposing it on specific data regions as done by the two previous kind of penalization. Anchored-Networks appears to be an efficient tool, for instance, in the detection of corrupted data (Ulmer et al., 2020), however, it is very hard to set the right anchors and trade-off parameter (Scalia et al., 2020). Large initial variance can lead to large weight variance but may not converge to accurate model in-distribution. The DARE approach is more practical as it starts to increase the weights after reaching an acceptable loss threshold which ensures to fit the training data.

## 4 EXPERIMENTS

The experiments are conducted on both regression and classification datasets. The source code of the experiments is available on GitHub[1]. We consider the following competitors: **Deep-Ensemble** (**DE**) (Lakshminarayanan et al., 2017), **NegCorr** (Shui et al., 2018), **AnchorNet** (Pearce et al., 2018), **MOD** (Jain et al., 2020) and **RDE** (D'Angelo & Fortuin, 2021). All are deep ensemble methods presented in Section 3. AnchorNet, NegCorr and MOD introduce a penalty term in the loss function multiplied by a trade-off parameter $\lambda$. The trade-off $\lambda$ and the anchor initialization parameter $\sigma$ for AnchorNet are selected through hold-out validation, as suggested in (Jain et al., 2020; Pearce et al., 2018). The kernel bandwidth for RDE is adaptive and chosen with the median heuristic as suggested in the corresponding work (D'Angelo & Fortuin, 2021). The validation loss is monitored across epochs. We restore the weights, at the training end, corresponding to the best validation loss epoch. For DARE, the parameter $\delta$ is set to $1/4$ and the weight saving strategy follows the monitoring process described in Section 2.2.

If nothing else is specified, the experiments are performed with ensemble of $M = 5$ fully-connected networks with 3 hidden layers of 100 neurons each and ReLU activations. The Adam optimization algorithm is used with learning rate 0.001 (Kingma & Ba, 2015). The batch size is chosen equal to 128. The experiments are repeated 5 times to compute standard deviations for the scores. For the regression experiments, we use the Gaussian NLL defined in (Lakshminarayanan et al., 2017) as loss function. Each network in the ensemble returns the 2-dimensional output $h_\theta(x) = (\mu_\theta(x), \sigma_\theta(x))$, for any $x \in \mathcal{X}$. The mean prediction of the ensemble $(h_{\theta^{(1)}}, ..., h_{\theta^{(M)}})$ is then equal to $\overline{\mu}(x) = (1/M) \sum_{m=1}^{M} \mu_{\theta^{(m)}}(x)$ and the prediction variance is computed through $\overline{\sigma}(x) = (1/M) \sum_{m=1}^{M} (\sigma_{\theta^{(m)}}(x)^2 + \mu_{\theta^{(m)}}(x)^2) - \overline{\mu}_\theta(x)^2$. For the classification experiments, the loss function is the NLL and a softmax activation is added at the end-layer of the networks, following common practice in the classification setting. However, for DARE, we observe that the softmax activation cancels the effect of increasing the weights. Indeed, the softmax activation inverses the correlation between large outputs and high uncertainty, resulting in over-confidence for OOD data. To avoid this negative effect, the loss function is set to the mean squared error, scaled by the number of classes: $\ell(h_\theta(x), y) = ||h_\theta(x) - Ky||_2^2$, for any $x, y \in \mathcal{X} \times \mathcal{Y}$, with $y$ the one-hot encoded label, $h_\theta(x)$ the predicted logit and $K$ the number of classes. To provide uncertainty quantification, we define the following "ad-hoc" uncertainty score:

$$u(x) = \frac{1}{M} \sum_{m=1}^{M} ||h_{\theta^{(m)}}(x) - K\widehat{y}_m||_2^2 + \frac{1}{M} \sum_{m=1}^{M} ||h_{\theta^{(m)}}(x) - \overline{h}(x)||_2^2. \tag{10}$$

Where $\widehat{y}_m$ is the one-hot encoding of the estimated label: $\arg\max_{k \in [|1,K|]} h_{\theta^{(m)}}(x)_k$ and $\overline{h}(x) = (1/M) \sum_{m=1}^{M} h_{\theta^{(m)}}(x)$. Equation (10) can be interpreted as the addition of the individual uncertainty estimation of each member to the ensemble prediction variance.

In the majority of previous works, OOD uncertainty quantification is studied in the perspective of OOD detection in the classification setting where examples from other classes / datasets are considered as OOD (D'Angelo & Fortuin, 2021; Lakshminarayanan et al., 2017; Liu et al., 2022; Van Amersfoort et al., 2020). For regression, few attempts of uncertainty quantification on shifted datasets have been conducted: (Jain et al., 2020) separates male and female faces for age prediction dataset and (Jain et al., 2020; Foong et al., 2019; Segonne et al., 2022) propose OOD version of several

---

[1]https://github.com/AnonymousAccount3/dare

UCI regression datasets (Dua & Graff, 2017). In this work, we propose a novel regression setup for uncertainty estimations on shifted domains based on the CityCam dataset (Zhang et al., 2017) which has been used in several domain adaptation regression experiments (de Mathelin et al., 2020; Zhao et al., 2018). Our setup models real-life domain shift scenarios where uncertainty quantification is challenged and offers a clear visual understanding of the domain shifts (cf Figure 3). For the classification experiments, we consider the OOD detection setup developed in (D'Angelo & Fortuin, 2021).

### 4.1 Synthetic experiments

We consider the "two-moons" binary classification dataset and the 1D regression experiment developed in (Jain et al., 2020). The visualization of the results is provided in Figure 1. The full description of the experiments is reported in Appendix.

We are interested in confirming the theoretical insights derived in Section 3, i.e. the weight variance is proportional to the training neuron activation variance and OOD data that activate neurons of small training activation variance propagate large values inside the network. Figure 2 presents the predicted uncertainty heat-map for one DARE network as well as the internal layer representations for the classification experiment. We observe that the OOD neuron activations grow from one layer to the next. A correspondence between features with low variance for the training data and large weights can be clearly observed. In the last hidden layer (layer 3), the OOD components are large in direction of low training variance (components 80 to 100) to which correspond large weights. This observation explains the large uncertainty score for the OOD example.

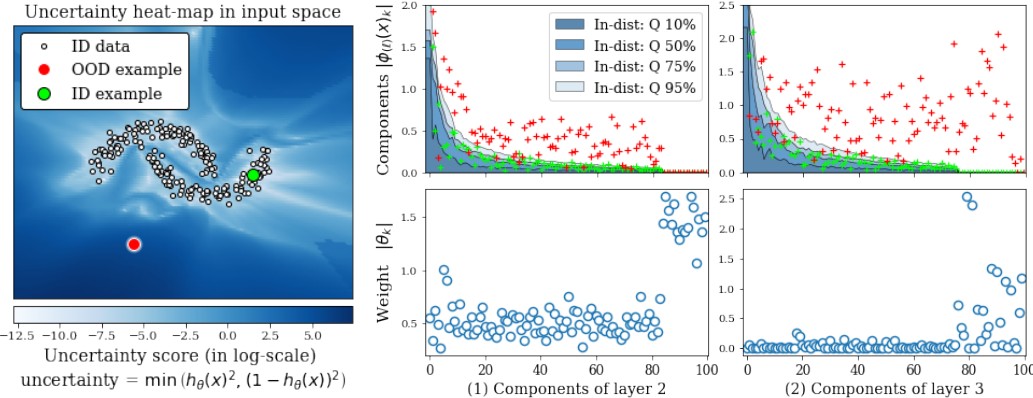

Figure 2: **Internal analysis of a DARE network** The uncertainty produced by one DARE member is presented on the left. On the right, the two figures on top present the expression of the training distribution in the three hidden layers (in blue) compared to the representation of one OOD example (in red). The components (neuron activations) are sorted in descending order of training variance. The two bottom figures present the average weight in front of each component, i.e. the mean weights that multiply the layer components to produce the next layer representation.

### 4.2 Regression experiments on CityCam

We propose, here, a novel regression setup for uncertainty estimations on shifted domains based on the CityCam dataset (Zhang et al., 2017) which has been used in several domain adaptation regression experiments (de Mathelin et al., 2020; Zhao et al., 2018). Our setup models real-life domain shift scenarios where uncertainty quantification is challenged and offers a clear visual understanding of the domain shifts (cf Figure 3). The CityCam dataset is composed of images coming from different traffic cameras. The task consists in counting the number of vehicles present on the image, which is useful, for instance, to control the traffic in the city. To get relevant features for the task, we use the features of the last layer of a ResNet50 (He et al., 2016) pretrained on ImageNet (Deng et al., 2009). We propose three different kinds of domain shift:

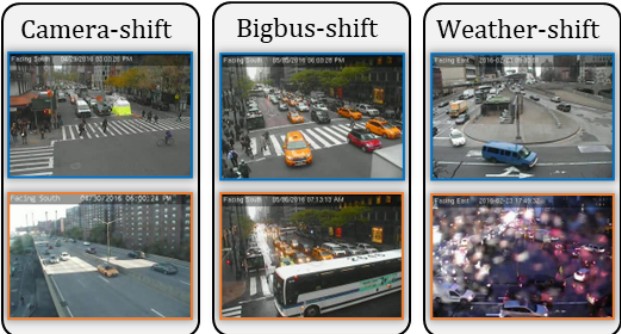

Figure 3: **CityCam experimental setups**. The top blue images correspond to in-distribution examples and bottom orange images to OOD examples.

**1. Camera-shift**: This experiment uses the images from 10 cameras in the CityCam dataset. For each round, 5 cameras are randomly selected as in-distribution while the 5 remaining cameras are considered as out-of-distribution.

**2. Bigbus-shift**: Images marked as "big-bus" referring to the fact that a bus appears and masks a significant part of the image (Zhang et al., 2017) are used to create the OOD dataset.

**3. Weather-shift**: Blurry images caused by water drops landed on the camera are used as OOD dataset.

These three experiments model real-life uncertainty quantification problems as the generalization of uncertainty estimates to unseen domains (camera-shift), the robustness to changes in data acquisition (weather-shift) and the detection of rare abnormal event (bigbus-shift). Further details on these experimental setups are provided in Appendix.

| Datasets / Methods | In-distribution Coverage | | | Out-of-distribution Coverage | | |
|---|---|---|---|---|---|---|
| | Camera | Bigbus | Weather | Camera | Bigbus | Weather |
| Deep Ensemble | 96.71 (0.54) | 97.60 (0.15) | 96.50 (0.15) | 63.0 (3.9) | 78.9 (0.9) | 88.8 (0.2) |
| Negative Correlation | 96.97 (1.34) | 97.68 (0.46) | 96.50 (1.37) | 63.8 (6.8) | 79.4 (2.9) | 89.0 (0.6) |
| MOD | 97.22 (1.05) | 97.82 (0.07) | 95.90 (0.15) | 65.6 (5.6) | 79.3 (0.0) | 88.8 (0.8) |
| Anchored Networks | 96.44 (0.32) | 96.72 (0.59) | 96.66 (0.30) | 64.1 (3.6) | 76.8 (2.2) | 89.8 (0.0) |
| RDE | 96.83 (0.13) | 97.19 (0.07) | 96.35 (0.61) | 64.0 (3.9) | 77.3 (0.4) | 89.2 (1.2) |
| DARE | 96.98 (0.16) | 96.55 (0.61) | 97.42 (0.15) | **70.9 (2.4)** | **80.0 (0.7)** | **93.7 (0.0)** |

Table 1: **In-distribution and Out-of-distribution Coverage for CityCam**. The coverage scores are reported after using conformal prediction on the validation dataset.

The number of epochs is set to 100 for Camera-shift and Bigbus-shift, and 200 for Weather-shift, based on the number of instances in the datasets. We notice that all methods fully converge before reaching this limit of epochs. To assess the ensemble quality for the regression experiments, we consider the "conformalized Out-of-distribution coverage". To compute this metric, the predicted standard deviations $\overline{\sigma}(x)$ are used to produce confidence intervals of level $1 - \alpha$, such that:

$$C(x, \beta) = \left[\overline{\mu}(x) + \Phi(\alpha/2)\overline{\sigma}(x) - \beta, \ \overline{\mu}(x) + \Phi(1 - \alpha/2)\overline{\sigma}(x) + \beta\right], \quad (11)$$

with $\Phi$ the cdf of the normal distribution and $\beta \leq 0$. The confidence intervals are then "conformalized" using conformal prediction Romano et al. (2019), the parameter $\beta \in \mathbb{R}$ is then defined such that a proportion $1 - \alpha$ of the validation data $(x, y)$ verify: $y \in C(x, \beta)$. We consider a confidence level $\alpha = 0.05$ and compute the coverage on the respective test and OOD datasets as follows:

$$\text{Cov}_{\text{test}} = \frac{1}{|\mathcal{S}_{\text{test}}|} \sum_{(x,y) \in \mathcal{S}_{\text{test}}} \mathbb{1}\left(y \in C(x, \beta)\right) \qquad \text{Cov}_{\text{ood}} = \frac{1}{|\mathcal{S}_{\text{ood}}|} \sum_{(x,y) \in \mathcal{S}_{\text{ood}}} \mathbb{1}\left(y \in C(x, \beta)\right) \quad (12)$$

The results are reported in Table 1. We first observe that the test coverage for all methods is very similar, as a consequence of the "conformalization" on the validation dataset which follows the same distribution as the test set. We observe, however, that DARE outperforms other uncertainty

quantification methods in terms of OOD coverage for the three experiments. in the camera-shift experiment, for instance, the Out-of-distribution coverage for DARE is more than 5 points above the second-best method.

## 4.3 Classification Experiments

| Methods | CIFAR10 | | | Fashion-MNIST | | |
|---|---|---|---|---|---|---|
| | SVHN | CIFAR100 | Accuracy | CIFAR10 | MNIST | Accuracy |
| DE (NLL) | 90.9 (0.4) | 86.4 (0.2) | **91.8 (0.1)** | 89.7 (0.9) | 62.7 (6.2) | **89.2 (0.2)** |
| AnchorNet | 91.0 (0.3) | **86.5 (0.2)** | **91.8 (0.0)** | 88.8 (1.1) | 68.7 (6.2) | 89.1 (0.2) |
| MOD | 91.3 (0.3) | 86.3 (0.3) | 91.7 (0.1) | 89.4 (1.7) | 60.8 (2.7) | 88.7 (0.4) |
| NegCorr | 91.3 (0.4) | 86.3 (0.4) | 91.7 (0.1) | 91.5 (0.8) | 68.9 (4.5) | 86.1 (0.6) |
| RDE | 91.2 (0.5) | 86.4 (0.3) | **91.8 (0.1)** | 90.1 (0.9) | 70.9 (5.8) | 89.1 (0.3) |
| DE (MSE) | 85.9 (1.2) | 77.7 (0.8) | 91.7 (0.1) | 96.5 (0.5) | 93.0 (5.3) | 88.6 (0.1) |
| DARE | **92.6 (0.7)** | 82.7 (0.5) | **91.8 (0.1)** | **97.7 (0.5)** | **97.4 (1.3)** | 87.2 (0.2) |

Table 2: **OOD detection results**. AUROC scores and ID accuracy are reported

We consider the experimental setup defines in (D'Angelo & Fortuin, 2021) for OOD detection on Fashion-MNIST and CIFAR10. The MNIST dataset is used as OOD dataset for Fashion-MNIST and the SVHN dataset for CIFAR10. We extend the experiments by adding CIFAR10 as OOD for Fashion-MNIST and CIFAR100 as OOD for CIFAR10. Thus, for both experiments, OOD detection is performed on both "Near-OOD" and "Far-OOD" datasets (Liu et al., 2022). To reduce the need of computational ressources for CIFAR10, we consider the "multi-head" setting (Lee et al., 2015), where deep ensemble of fully-connected networks are trained over the penultimate layer of a pretrained ResNet32 (He et al., 2016).

The obtained results are reported in Table 2 for DARE and the competitors. To fully evaluate the impact of the DARE optimization, we add the results obtained with a Deep Ensemble trained with the mean squared error (DE (MSE)) which is equivalent to a DARE with $\lambda = 0$. We train 5 networks in each ensemble and repeat the experiments 5 times. The AUROC metric is used, computed with the uncertainty score defined in Equation (10) for DARE and DE (MSE) and the entropy for the other methods. We observe that DARE globally improves the OOD detection. For instance, in Fashion-MNSIT, we observe an improvement of 8 points on CIFAR10 and 34 points on MNIST compared to DE, with a lost of only 2 points of in-distribution accuracy. Mild results are obtained For CIFAR10, we observe an improvement for SVHN but not for CIFAR100. We argue that it is mainly due to the use of the mean squared error, which is not suited for this experiment, as indicated by the poor results of DE (MSE). Notice that for the Fashion-MNIST experiment, the contrary is observed, as DE (MSE) provides an important improvement. We finally underline that, DARE always performs better than its DE (MSE) counterpart.

## 5 Limitations and Perspectives

For now, the efficiency of DARE is limited to fully-connected neural network with piece-wise linear activation and linear end-activation. Moreover, the threshold setting is still based on a heuristic, which may be suboptimal. We have seen, however, that DARE can benefit to a final fully-connected network head placed on top of deep features obtained with convolutions. Thanks to the practical aspect of DARE, the method can be combined to other deep ensemble or OOD detection methods. One can use a specific training process and then apply DARE afterward to increase diversity. Future work will consider a "Bayesian" version of DARE by adding Gaussian noise with increasing variance to the weights of pretrained networks.

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

# A  PROOF OF THEOREM 1

We first remind the assumptions and notations used for the Theorem:

**Notations**:

- $X \in \mathbb{R}^{n \times p}$ and $y \in \mathbb{R}^n$ are respectively the input and output data which compose the training set $\mathcal{S}$.
- $\mathcal{H} = \{x \to x^T \theta; \theta \in \mathbb{R}^p\}$ is the ensemble of linear networks.
- $\mathcal{L}_{\mathcal{S}}(h_\theta) = \frac{1}{n}||X\theta - y||_2^2$ is the mean squared error.
- $s^2 = (s_1^2, ..., s_p^2) \in {\mathbb{R}_+^*}^p$ is the diagonal of the matrix $\frac{1}{n}X^T X$.
- $\Theta_{\sigma^2}$ is the weights distribution of the anti-regularized ensemble $\mathcal{H}_\tau \subset \mathcal{H}$, with $\sigma^2$ the weights variance.

**Assumptions**:

$$\exists \theta^* \in \mathbb{R}^p; X\theta^* = y \tag{13}$$

For any $h_\theta \in \mathcal{H}_\tau$:

$$\theta \sim \Theta_{\sigma^2}; \ \mathbb{E}[\theta] = \theta^*, \ \mathrm{Cov}(\theta) = \mathrm{diag}(\sigma^2) \tag{14}$$

$$\mathbb{E}[\mathcal{L}_{\mathcal{S}}(h_\theta)] \leq \delta\tau \tag{15}$$

Where $\delta > 0$ and $\mathrm{diag}(\sigma^2)$ is the diagonal matrix of values $\sigma^2 \in \mathbb{R}_+^p$ verifying:

$$\sigma^2 = \operatorname*{arg\,max}_{\sigma^2 = (\sigma_1^2, ..., \sigma_p^2)} \sum_{k=1}^p \log\left(\theta_k^{*2} + \sigma_k^2\right) \tag{16}$$

**Theorem 2.** *There exist a constant $C > 0$ such that for any $k \in [|1, p|]$, the variance of the $k^{th}$ weight component is expressed as follows:*

$$\sigma_k^2 = \max\left[\frac{C\delta\tau}{s_k^2} - \theta_k^{*2}, 0\right] \tag{17}$$

*Proof.* Let's introduce the variable $z \sim \mathcal{N}(0, \mathrm{diag}(\sigma^2))$, verifying: $\theta = \theta^* + z$. From Assumption 15, we derive that:

$$\begin{aligned}
\mathcal{L}_{\mathcal{S}}(h_\theta) &= \frac{1}{n}||X\theta - y||_2^2 \\
&= \frac{1}{n}||X(\theta^* + z) - y||_2^2 \\
&= \frac{1}{n}||X\theta^* - y + Xz||_2^2 \\
&= \frac{1}{n}||Xz||_2^2 \ \text{ (by definition of } \theta^*) \\
&= \frac{1}{n}\sum_{i=1}^n \sum_{k=1}^p \sum_{j=1}^p X_{ik} X_{ij} z_k z_j
\end{aligned} \tag{18}$$

Thus, we have:

$$\begin{aligned}
\mathbb{E}[\mathcal{L}_{\mathcal{S}}(h_\theta)] &= \frac{1}{n}\sum_{i=1}^n \sum_{k=1}^p \sum_{j=1}^p X_{ik} X_{ij} \mathbb{E}[z_k z_j] \\
&= \frac{1}{n}\sum_{i=1}^n \sum_{k=1}^p X_{ik}^2 \sigma_k^2 \ \text{ (by definition of } z) \\
&= \sum_{k=1}^p \left(\frac{1}{n}\sum_{i=1}^n X_{ik}^2\right) \sigma_k^2 \\
&= \sum_{k=1}^p s_k^2 \sigma_k^2
\end{aligned} \tag{19}$$

Combining this results with Assumption 16, we show that $\sigma^2$ verifies:

$$\underset{\sigma^2 \in \mathbb{R}_+^p}{\text{maximize}} \quad \sum_{k=1}^{p} \log(\sigma_k^2 + \theta_k^{*2})$$

$$\text{subject to} \quad \sum_{k=1}^{p} s_k^2 \sigma_k^2 \leq \delta\tau \tag{20}$$

This expression is a "water-filling" problem (cf Boyd et al. (2006) Example 5.2) with weighted constraint. The inequality constraint $\sum_{k=1}^{p} s_k^2 \sigma_k^2 \leq \delta\tau$ can be written as an equality constraint, as any increase of $\sigma_k^2$ induces an increase of the objective function.

To solve this problem, we introduce the Lagrange multipliers $\mu \in \mathbb{R}_+^p$ for the constraints $\sigma^2 \geq 0$ and the multiplier $\alpha \in \mathbb{R}$ for the constraint $\sum_{k=1}^{p} s_k^2 \sigma_k^2 = \delta\tau$. By considering the Lagrangian as a function of $\sigma^2$, the KKT conditions are then written:

$$\frac{-1}{\theta_k^{*2} + \sigma_k^2} + \alpha s_k^2 - \mu_k = 0 \ \forall\, k \in [|1,p|] \tag{21}$$

$$\sigma_k^2 \geq 0, \ \mu_k \geq 0, \ \mu_k \sigma_k^2 = 0 \ \forall k \in [|1,p|] \ \text{ and } \ \sum_{k=1}^{p} s_k^2 \sigma_k^2 = \delta\tau \tag{22}$$

Leading to:

$$\sigma_k^2 \left( \frac{-1}{\theta_k^{*2} + \sigma_k^2} + \alpha s_k^2 \right) = 0 \ \forall\, k \in [|1,p|] \tag{23}$$

$$\sigma_k^2 \geq 0, \ \alpha s_k^2 \geq \frac{1}{\theta_k^{*2} + \sigma_k^2}, \ \forall k \in [|1,p|] \ \text{ and } \ \sum_{k=1}^{p} s_k^2 \sigma_k^2 = \delta\tau \tag{24}$$

Then, for any $k \in [|1,p|]$, we have:

$$\sigma_k^2 = 0 \ \text{ or } \ \sigma^2 = \frac{1}{\alpha s_k^2} - \theta_k^{*2} \tag{25}$$

As $\sigma_k^2 \geq 0$, we deduce that:

$$\sigma_k^2 = \max\left[ \frac{1}{\alpha s_k^2} - \theta_k^{*2}, 0 \right] \tag{26}$$

With $\alpha$ verifying:

$$\sum_{k=1}^{p} s_k^2 \max\left[ \frac{1}{\alpha s_k^2} - \theta_k^{*2}, 0 \right] = \delta\tau$$

$$\sum_{k=1}^{p} \max\left[ \frac{1}{\alpha\delta\tau} - \frac{s_k^2 \theta_k^{*2}}{\delta\tau}, 0 \right] = 1 \tag{27}$$

By defining $C = \frac{1}{\alpha\delta\tau}$, we have:

$$\sum_{k=1}^{p} \max\left[ C - \frac{s_k^2 \theta_k^{*2}}{\delta\tau}, 0 \right] = 1 \tag{28}$$

Which imposes $C > 0$.

We conclude then, that there exists $C > 0$ such that, for any $k \in [|1,p|]$:

$$\sigma_k^2 = \max\left[ \frac{C\delta\tau}{s_k^2} - \theta_k^{*2}, 0 \right] \tag{29}$$

$\square$

## B  PROOF OF COROLLARY 1

**Corollary 2.1.** *Let $\mathcal{H}_\tau$ be the large weights ensemble defined by Assumptions 14, 15, 16 and $x \in \mathbb{R}^p$ an input data, the variance of prediction for $h_\theta \in \mathcal{H}_\tau$ is:*

$$\underset{\theta \sim \Theta_{\sigma^2}}{\mathrm{Var}} \left[ h_\theta(x) \right] = \sum_{k=1}^{p} x_k^2 \max \left[ \frac{C\,\delta\,\tau}{s_k^2} - \theta_k^{*2}, 0 \right] \tag{30}$$

*Proof.* Let's consider $x \in \mathbb{R}^p$ and $h_\theta \in \mathcal{H}_\tau$:

$$\begin{aligned}
\underset{\theta \sim \Theta_{\sigma^2}}{\mathrm{Var}} \left[ h_\theta(x) \right] &= \underset{\theta \sim \Theta_{\sigma^2}}{\mathrm{Var}} \left[ \sum_{k=1}^{p} x_k \theta_k \right] \\
&= \sum_{k=1}^{p} x_k^2 \sigma_k^2 \quad \text{(by Assumption 14)}
\end{aligned} \tag{31}$$

$\square$

## C  SYNTHETIC EXPERIMENTS

We consider the two synthetic experiments : Two moons classification and 1D regression :

- **Two Moons Classification** : 200 random data points are generated from the two moons generator[2] to form the training set. We consider the three uncertainty quantification methods: Deep Ensemble (NLL), Deep Ensemble (MSE) and Deep Anti-Regularized Ensemble (DARE). The first method implements a softmax activation function at the end layer and use the multi-classication NLL as loss function. The two others use a linear activation at the end layer and the mean squared error as loss function. A fully-connected network of three layers with 100 neurons each and ReLU activations is used as base network. Each ensemble is composed of 20 networks. The Adam Kingma & Ba (2015) optimization is used with learning rate 0.001, batch size 32 and 500 epochs. For DARE, the threshold is set to $\tau = 0.001$. To compute the uncertainty scores, the entropy metric Lakshminarayanan et al. (2017) is used for Deep Ensemble (NLL) while Deep Ensemble (MSE) and DARE use the uncertainty metric for classification with MSE loss function. We report the uncertainty map produced by each method in Figure 4. We also report the results for the OOD detection task in Figure 5. To produce this last results, we sample 50 random points from the two moons generator which acts as validation data. We compute the uncertainty scores on the validation set and use the 95% percentiles as threshold for OOD detection. Any data points below this threshold are considered as in-distribution data (in white) and the others as OOD (in dark blue).

- **1D Regression** : We reproduce the synthetic univariate Regression experiment from Jain & Grauman (2016). We consider the three methods: Deep Ensemble (NLL), Deep Ensemble (MSE) DARE. We use the gaussian NLL loss for regression defined in Lakshminarayanan et al. (2017) for Deep Ensemble (NLL) and DARE. The mean squared error is used as loss function for Deep Ensemble (MSE). The threshold for DARE is set to $\tau = 0.1$. For any data $x \in \mathbb{R}$, each method, return two values $\mu_x$ and $\sigma_x$ (cf Lakshminarayanan et al. (2017)). We report in Figure 6 the confidence intervals $[\mu_x - 2\sigma_x, \mu_x + 2\sigma_x]$ (in light blue) for any $x$ in the input range.

---

[2]https://scikit-learn.org/stable/modules/generated/sklearn.datasets.make_moons.html

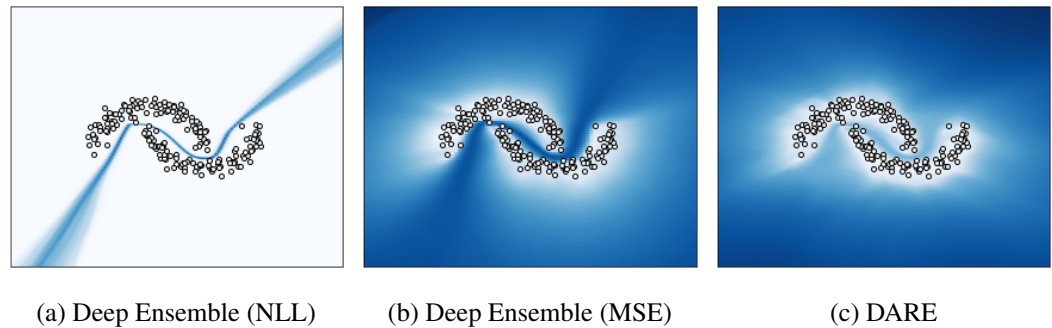

(a) Deep Ensemble (NLL)     (b) Deep Ensemble (MSE)     (c) DARE

Figure 4: **Two-moons uncertainty estimation** Darker areas correspond to higher predicted uncertainty.

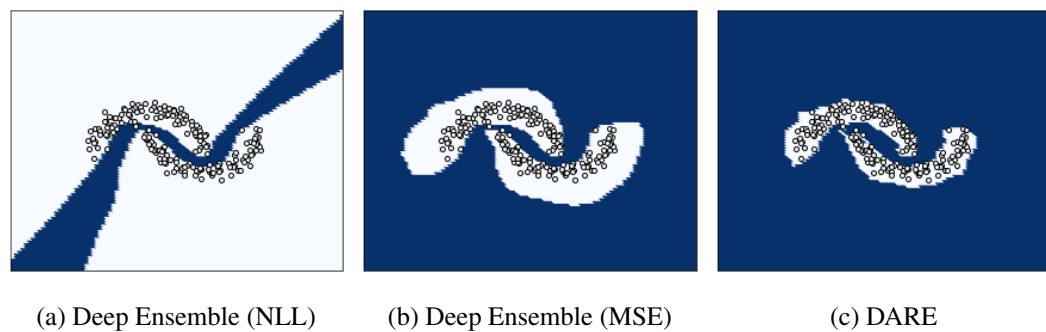

(a) Deep Ensemble (NLL)     (b) Deep Ensemble (MSE)     (c) DARE

Figure 5: **OOD detection**. Data classified as OOD are in dark blue OOD while the ones classified as in-distribution are in white.

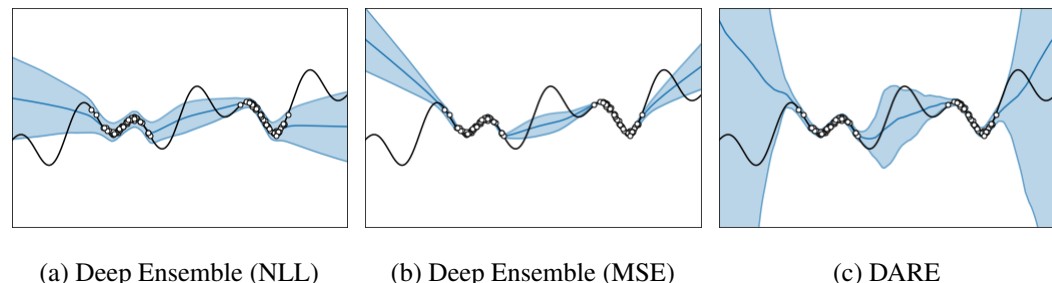

(a) Deep Ensemble (NLL)     (b) Deep Ensemble (MSE)     (c) DARE

Figure 6: **1D Regression uncertainty estimation**. the confidence intervals for $\pm 2\sigma$ are given in light blue.

## D  CITYCAM EXPERIMENTS

We consider the three following experiments:

- **Weather-shift**: in this experiment, we select the images of the three cameras n°164, 166 and 572 and use the images recorded during February the $23^{\text{th}}$. During this particular day, the weather significantly change between the beginning and the end of the day. Thus, to evaluate the robustness of the method to a change in the weather condition, we split the dataset in two subsets: we consider the images recorded before 2 pm as in-distribution and the others as out-of-distribution. we observe a shift in the weather condition around 4 pm which produces a visual shift in the images. After 4 pm, the rain starts to fall and progressively damages

the visual rendering of the images due to the reverberation. This is particularly the case for camera n°164, as the water drops have landed on the camera and blur the images. Each dataset is composed of around 2500 images.

- **Camera-shift**: this experiment uses the images from 10 cameras in the CityCam dataset. For each round, 5 cameras are randomly selected as in-distribution while the 5 remaining cameras are considered as out-of-distribution. In average, each set is composed of around 20000 images.

- **Bigbus-shift**: The CityCam dataset contains images marked as "big-bus" referring to the fact that a bus appears and masks an significant part of the image Zhang et al. (2017). We then select the 5 cameras for which some images are marked as "big-bus" and use this marker to split the dataset between in-distribution and out-of-distribution samples. The in-distribution set is composed of around 17000 images while the other set contains around 1000 images.

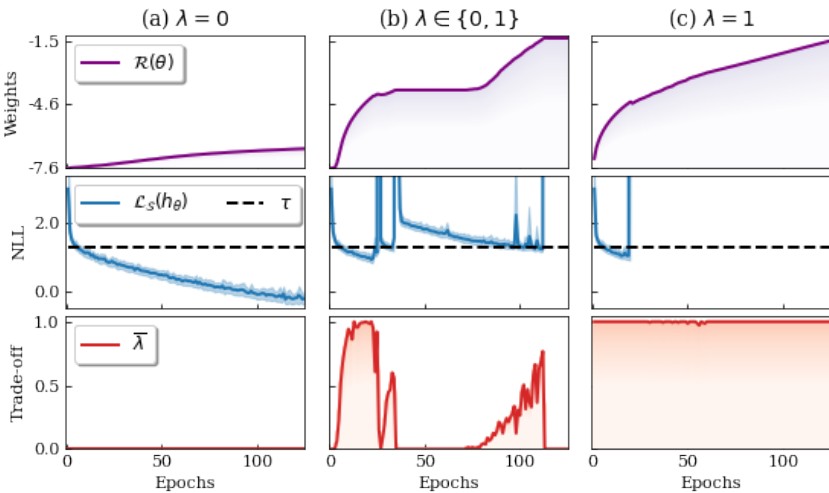

Figure 7: **Training behaviors for different $\lambda$ settings**. Evolution of the weights, the training loss and the average trade-off parameter over the epochs ((a) $\equiv$ DE and (b) $\equiv$ DARE).

The impact of the $\lambda$ control in the DARE optimization is presented in Figure 7, we observe that the setting $\lambda = 0$ (equivalent to DE) leads to a small weight increase whereas the setting $\lambda = 1$ leads to a loss explosion. The control setting $\lambda \in \{0, 1\}$ allows the weight increase while maintaining the loss above $\tau$.

