# OpenReview forum: "Deep Anti-Regularized Ensembles"
_ICLR.cc/2024/Conference — Submitted to ICLR 2024_

### Official Review · Reviewer_vXUV · 2023-10-27

**Soundness:** 3 good
**Presentation:** 2 fair
**Contribution:** 3 good
**Rating:** 5
**Confidence:** 3

**Summary:**

The authors propose a method of regularizing ensembles which flips the sign of the normal L2 regularization term. The goal is to create a model with large weights, where an OOD sample will produce extreme values, while the ID samples from the training set are not negatively impacted by the large weights.

**Strengths:**

- The method is novel and intuitive.
- It immediately makes sense to the reader as to why it should work, and it appears simpler and more 'parallelizable' when compared to other ensemble diversity methods.

**Weaknesses:**

- Abstract: I think you should introduce the concept of regularization reducing weights before stating the term "large weights network?" because at this point, the reader has no way of knowing the meaning of this statement until reading further.

- If you propose setting the tau validation loss parameter based on the validation loss of a trained model, then the method requires a complete training of a model from scratch before even beginning to train the proposed model. By definition, this training cannot be done in parallel.

- Below Equation 6, the text refers to "Assumption 4" which has not been properly defined. If the equations are meant to be assumptions, then they should be properly labeled as such.

- If $s^2$ is supposed to denote the input feature variance as stated before equation 7, and implied in the $X^\top X$ before equation 4, then the inputs features should be defined as being mean centered.

- Before Corollary 1.1, the following is said: "We see the importance of choosing a strictly concave function like the logarithm..." Where do we see this? Nothing has been demonstrated up until this point which would lead to this conclusion. If this is an insight gained from the specific details of the proof (which I had not read at this point in the paper), then something more useful should be said here.

- Before equation ten, the following statement is made: "To avoid this negative effect, the loss function is set to the mean squared error, scaled by the numberof classes." Could you be more specific about why this choice was made? Why should one scale the one hot vector with the number of classes? Wouldn't this mean that for a very high number of classes, the high magnitude outputs of a anti regualized ensemble would become hard to distinguish since K is also a high value?

- What are the red and green markers on the top row of plots in figure 2? They do not have a label.


- The presentation in Table 2 is confusing to read. It is not immediately clear whaty the accuracy and dataset columns are showing. One must deduce that for instance the SVHN, CIFAR100, and Accuracy columns under CIFAR10 are showing OOD Detection AUROC for SVHN and CIFAR100 and Accuracy on CIFAR10, is this correct?


- In the last paragraph before section 5: "We train 5 networks in each ensemble and repeat the experiments 5 times." If those are the only models which are trained, then how do you learn the $\tau$ parameter?

**Questions:**

- Section 2.2: Is the network really trained with batch gradient descent? Or mini-batch gradient descent?
- It seems like ReLU would not be ideal for this application. The reason why I say this is that ReLU's can "die" and become permanently deactivated once they reach a zero value and receive no gradient. For this reason, I would think that a smoother activation function (such as GELU) may allow for more large weights in the model which would otherwise not be receiving any gradient in a ReLU network.

---

Overall, I like the work, but my main concern lies with:

- some details around the presentation, which can be cleared up in a rebuttal
- the classification setting, using the mean squared error with the one hot vector makes me feel skeptical. I would not expect a network trained with this one hot MSE to perform as well as a softmax, and yet it seems to perform just as well as the baselines which do use a softmax. Can you provide any further ablation on this loss function or references to prior works which use a similar form of loss function and achieve results comparable to a softmax?

---

> ### Comment · Reviewer_vXUV · 2023-11-30
> **Thank you for the response.**
>
> Thank you for the response to my review and performing more classification experiments. Based on the response from the authors, I will keep my initial score.
>
> I do still think this is a novel approach which has great potential.

---

### Official Review · Reviewer_d3RB · 2023-10-30

**Soundness:** 2 fair
**Presentation:** 3 good
**Contribution:** 3 good
**Rating:** 5
**Confidence:** 4

**Summary:**

This work proposes a method called “deep anti-regularized ensembles”. It proposes a procedure that minimizes loss whilst also maximizing the magnitude of the weights within each member of an ensemble. This constrained optimization is achieved using a control process where the auxiliary weight maximization term is only included when training loss is below a certain threshold. It is argued that an ensemble consisting of networks with larger weights associated with neurons that activate more frequently on OOD data will result in reduced overconfidence in OOD predictions for the aggregate ensemble. Experiments are performed on both regression and classification datasets.

**Strengths:**

* The idea of increasing diversity by maximizing the magnitude of weights whilst maintaining the train performance is interesting and creative. I also appreciate the simplicity of the core idea.
* The control process for balancing these objectives seems like a good approach to solving this constrained optimization problem.
* The coverage of related work is reasonable.
* The application of making machine learning methods more robust to out-of-distribution data shifts is important and relevant.
* The paper is generally well written and clear to follow (although I have some issues with Sec 2.3 - see weaknesses).
* Code is provided to reproduce experiments (although I have not tested the code, just verified that it looks reasonably usable).

**Weaknesses:**

* **Theoretical Analysis in Sec 2.3** - I found this section to be very unclear relative to the rest of the paper. While I think it is possible that there is an interesting motivation in this line of reasoning, I found it unconvincing in its current form. I would highly recommend that the authors re-write this section entirely to make their argument much more clear. I list some of the issues I encountered below:
    * The claim that this setting offers valuable insights into the behavior between layers in a neural network seems to not necessarily be true. Anything proved in this section and the intuitions developed may not directly carry over due to important differences (e.g. linear regression to a target optimizes a different loss than between two hidden layers of a neural network, this section seems to not account for interaction terms which would surely exist in a neural network, …). I would suggest that linear regression could be valuable in itself without needing to claim that it is a good model for hidden layers of a neural network.
    * Eqns (4), (5), and (6) were quite unclear to me. Are these purely assumptions/definitions? If so it might be better to use the correct notation “:=“. Particularly eqn (6) is unclear, would the solution to this arg max not simply be $\sigma = \infty$ without further assumptions? Are we maximizing the elements of the vector? The problem setup and message that the authors wish to communicate is unclear.
    * The connection to theorem 1 is not explicit. Are the authors saying that because eqn (6) takes its current form this means it is optimal in some way? It is not clear to me what is being shown here.
    * In this section and throughout the rest of the paper I would recommend being more clear in distinguishing between scalars and vectors by using bold font for the latter and for matrices.
    * Is the expectation over the training data?
    * On line 5 of paragraph 2, why is there an $n$ term multiplied by the loss function?
    * The statement $\theta$ centered in $\theta^*$. What does this mean? Or do the authors mean “centered at”?
    * Typo: “loosing in generality” -> “loss of generality”.


* **Experimental design** - I found that the experiments contained too many moving parts and resulted in it being difficult to conclude that the method was sufficiently isolated. I would have liked to have seen much more stripped-back experiments that provide clear ablated evidence of the authors' claims around their method. Furthermore, I would have liked to have seen more evidence around the claims that (a) some neurons activate infrequently on training data but more frequently on OOD data, (b) the weights associated with these neurons grow more than others, and (c) this results in higher uncertainty without simply resulting in degenerate predictions. I note a number of additional components that were at play in the experiments below.
    * The conformalization procedure used (page 8) adjusts the intervals produced by the authors' method by using a calibration set to guarantee marginal coverage in distribution. Since conformal quantile regression simply uses the provided intervals as an initialisation for its procedure it would seem that the performance gains could be due the the conformalization rather than the authors' method. It is unclear why the authors' relationship would make a particularly good initialization for this method and this is not discussed.
    * The pretrained ResNet used in the classification experiments would seem to undermine the random initialization aspect of ensembling upon which the method claimed to rely. Are the weights frozen or trainable?
    * Using multiple heads on a single base model seems to make this no longer an ensembling approach and also contradicts some of the ensemble-based motivation earlier in the paper.
    * The experiments seem to generally evaluate some form of coverage on OOD data. However, it is possible that this could be achieved by causing each ensemble member to perform degenerately but independently on OOD data. This may result in high variance predictions but would seem not to be useful in practice. I think these experiments need to provide some measure of error on OOD or at least in distribution test set data to ensure this is not the case.
    * The fact that a custom loss function was required and that the more standard softmax was found to be ineffective.
    * The inclusion of the $K$ term in this loss function seems to be another introduced heuristic.

* **Related works** - The authors seem to have missed [1] which introduced a very similar approach to the cited work of Pagliardini et. al. I would also suggest that [2] should be cited as an even earlier work that made an explicit connection between ensemble performance and diversity before the works cited already.

* **Custom loss function** - If we remove the $K$ term and replace the addition with a subtraction, the custom loss function proposed in eqn (10) appears to be exactly the multiple output MSE of the entire ensemble decomposed into the individual losses and the diversity between them as originally derived in [2]. Furthermore, it appears by adding (rather than subtracting) the second term the authors are, in fact, _minimizing_ diversity rather than encouraging it as it appears they intended.  This may actually explain why it performs more favorably as recent work in [3,4] have investigated this joint objective in the case of deep ensembles and found that discouraging diversity in this way results in better performance. Could the authors comment on the apparent contradiction between their penalisation of diversity in this objective and their goal of encouraging diversity on OOD data?

* **Name of method** - While I appreciate that regularisation is often performed by shrinking or sparsifying network weights, the more general idea of regularisation is of reducing the size of the hypothesis space using some inductive bias. In my view, the proposed method is doing exactly that by seeking solutions that maximize the magnitude of weights associated with neurons that activate more frequently on OOD data. Therefore, I would suggest that naming this method as “anti-regularisation” may be somewhat of a misnomer and doesn’t accurately reflect what the method actually does. A more accurate name would be something closer to “anti-shrinkage”.

[1] Lee, Y., Yao, H., & Finn, C. (2022, September). Diversify and disambiguate: Out-of-distribution robustness via disagreement. In The Eleventh International Conference on Learning Representations.

[2] Krogh, A., & Vedelsby, J. (1994). Neural network ensembles, cross validation, and active learning. Advances in neural information processing systems, 7.

[3] Abe, T., Buchanan, E. K., Pleiss, G., & Cunningham, J. P. (2023). Pathologies of Predictive Diversity in Deep Ensembles. arXiv preprint arXiv:2302.00704.

[4] Jeffares, A., Liu, T., Crabbé, J., & van der Schaar, M. (2023). Joint Training of Deep Ensembles Fails Due to Learner Collusion. arXiv preprint arXiv:2301.11323.

**Questions:**

I found the control process of optimizing $\tau$ to be a sensible approach. I’m curious if the authors also tried other approaches (e.g. post hoc fine-tuning of a regular ensemble with eqn (1)) or if their intuition led them to this approach?

In the related works section the authors discuss three categories of approaches for penalizing for diversity in deep ensembles. I think it would be useful to include some high-level discussion on why the authors believe the third of these categories to be more effective than the others. I don’t necessarily disagree with the authors on this point, but I think it would be instructive for readers to better understand the authors' perspective on why and when different methods will excel. Could the authors provide some discussion on the strengths and weaknesses of these categories (even if only in the appendix)?

In the section on parallel optimization, the authors claim that “each network in the ensemble is trained […] independently of others”. However, it appears that both the classification objective in eqn (10) and the regression objective contain a term that is a function of the entire ensemble (i.e. $\bar{h}(x)$). Backpropagating through this would require the entire ensemble which would therefore prevent parallel optimisation. Does this method in fact not use parallel optimization or is something like a “stop gradient” applied?

---

### Official Review · Reviewer_i1WL · 2023-10-30

**Soundness:** 3 good
**Presentation:** 3 good
**Contribution:** 3 good
**Rating:** 5
**Confidence:** 3

**Summary:**

The paper proposes a new way (DARE) for encouraging diversity (for out-of-distribution data) when training deep ensemble: It uses a "anti-regularization" term to encourage the weights of networks to have a large value. Of course, this would sacrifice the in-distribution predictive performance, therefore the authors further propose a thresholding mechanism to ensure the in-distribution accuracy doesn't drop. The author provides theoretical results showing on why this method works and provides empirical experiment results on toy dataset and real-world image datasets, showing that the proposed method has improved OOD detection performance and uncertainty quantification ability. Overall, I think the idea is interesting, but the experiment results are not very convincing, in that: 1. The experiments only consider fully-connected layers model (or only performing ensemble over the last linear layers in the CNN). 2. Although the OOD detection performance increased, the accuracy dropped (when in-distribution data is fashion MNIST).

**Strengths:**

- The proposed method is well-motivated and novel. Although it sounds like a heuristic, it actually has some theories supported.

- The results on the toy dataset look very promising: It shows uncertainty quantification behavior as good as distance-aware models, such as Gaussian processes.

- Experiment on real-world application: CityCam, demonstrates the effectiveness of the method in real-world distribution shift challenges.

- The experiment settings are carefully discussed, allowing better reproducibility.

**Weaknesses:**

- In networks with normalization layers, e.g. BatchNorm, the network would become scale-invariance, then increasing the weight in intermediate layers would not change the output. The proposed method could therefore be less effective.

- The paper only conducts experiments with fully-connected layer models. I could not find discussions in the manuscript for this choice.

- The proposed method requires using MSE loss for classification, which is not a typical choice.

- The empirical results are not appealing enough: In table 2, when the in-distribution data is CIFAR-10, DARE has better AUROC in SVHN but worse AUROC on CIFAR100 compared with baseline approaches. Therefore it is unclear to me whether one should use DARE for problems of CIFAR-10 level complexity.

- The presentation of the results in table 2 is not very clear: If I understand correctly, the 1,2 and 4, 5 columns are AUROC and 3,6 columns are for accuracy? This should be revealed more clearly in the table caption.

**Questions:**

- Why does the left-most uncertainty subfigure in Figure 2 look different from Figure 1(a)? Are they based on the same results but just different in scaling?

- What makes DARE perform better than the baseline approaches? Is it because it trains the ensemble components independently?

- The idea of adjusting the weight's variance according to the features' covariance structure is similar to method used in [1, 2], but [1, 2] propose to reduce the variance on the "OOD directions" to improve robustness on OOD samples. Nevertheless, I think it is worth discussing them in the related work section.

[1] Izmailov, Pavel, et al. "Dangers of Bayesian model averaging under covariate shift." Advances in Neural Information Processing Systems 34 (2021): 3309-3322.

[2] Wang, Xi, and Laurence Aitchison. "Robustness to corruption in pre-trained Bayesian neural networks." The Eleventh International Conference on Learning Representations. 2022.

---

### Official Review · Reviewer_jYun · 2023-11-04

**Soundness:** 3 good
**Presentation:** 3 good
**Contribution:** 2 fair
**Rating:** 3
**Confidence:** 4

**Summary:**

The paper focuses on the problem of improving uncertainty quantification of deep ensembles, proposing the inclusion of an "anti-regularization" term that pushes the weight values to be larger instead of smaller. Specifically, the loss is $L(\theta) = L_S(h_\theta) - \lambda R(\theta)$, where $R(\theta) = 1/d \sum_{k=1}^d \log(\theta_k^2)$. The insight is that an ensemble of models with large weight variance would produce large output variance for any data points, and in order to make accurate predictions on the training distribution while retaining good uncertainty calibration OOD, the output variance needs to be reduced for those training points while being kept large elsewhere. The paper gives some theoretical motivation, and then provides a few experiments showing the promise of the method.

**Strengths:**

The paper does a good job of identifying and conveying an insight (larger weight variance leads to larger output variance), providing theoretical motivation, and then running experiments to test out that method.

**Weaknesses:**

The main weaknesses of this paper are the limited set of experiments and mixed results. ResNet-32 on CIFAR-10 with only the final layer ensembled provides limited data. It's a fine experiment to include in the paper, but a larger-scale experiment with a fully-ensembled bigger model on a bigger dataset (even on ImageNet, which is not that large anymore) would provide much more information on the viability of this approach and whether it provides enough benefit. Concretely, looking at Table 2, both in-distribution accuracy and OOD detection is mixed for DARE, including worse results for the important near-OOD case of CIFAR-100 vs CIFAR-10. As noted in the limitations section, this paper is also missing comparisons to Bayesian neural nets (rank-1 BNNs and Prior networks could be nice experimental comparisons here). In the current environment, experiments with language models would also be extremely useful. Overall, the experimental setup is limited and the results are not that compelling in their current state.

**Questions:**

As noted in the weaknesses above, I would suggest running larger experiments to better allow the reader to understand the efficacy of the approach. Uncertainty quantification is an important problem, so I encourage continued work here!

---

### Author Response · Authors · 2023-11-23
**General Response to Reviewers**

The authors would first like to thank all reviewers for their high-quality reviews. We are indeed delighted to see such detailed and relevant comments, which demonstrate the reviewers' deep understanding of the proposed approach. It will certainly help us to improve our work.

First of all, we are pleased to note that all reviewers found the "large weights penalization" to be original, simple and promising. However, as they pointed out, there are still many issues to be resolved. We identify the following four points as the reviewers' main concerns:

1) The results for the classification experiments are not good enough. The setup with multiple fully-connected heads and MSE loss is not convincing and somewhat confusing.

2) The term "anti-regularization" is not appropriate to describe the method, as imposing large weights reduces the size of the hypothesis space, which is a form of regularization.

3) Experiments are all conducted with fully-connected networks with ReLU activation. The reviewers would like to know how DARE interacts with other network architectures, such as CNN or other activation functions as GELU, Leaky-ReLU...

4) The theoretical part lacks clarity. Assumptions / definitions are not clearly stated as such. The definition of sigma is confusing, as it implies that sigma = infinity.

Here are the authors' response to these four points:

1) As stated in the paper, an important limitation of DARE is that it doesn't work well with the softmax activation, which can be "saturated". We then use the MSE loss, which, indeed, is not suited for classification. We have conducted additional classification experiments, following the "posthoc" setup of the OpenOOD benchmark [1] on both datasets CIFAR10 and CIFAR100. We use the pretrained ResNet18 networks to provide input features and then train DARE on it. We obtain strong results for CIFAR10 (DARE appears among the top 3 "post-hoc" methods). However, we observe poor results for CIFAR100 (DARE provides worse results than the Deep Ensemble trained on the extracted features). We suspect DARE to be sensitive to the number of classes (or to the difficulty of the task). This issue may come from the use of the MSE loss, but we didn't find another classification loss suited for DARE. From these observations, we conclude that some work is still necessary to provide a relevant version of DARE for classification.

2) We agree on the fact that "anti-regularization" is not appropriate. We will work on finding another term to describe the method.

3) We have tried to use DARE with other network architectures, as the entire ResNet18, and observed that the method mainly works for fully-connected networks. As reported by the reviewers, DARE may not work well with BatchNorm. To overcome this issue, we apply DARE to fine-tune pretrained ensembles of networks and freeze the BatchNorm parameters (which are already trained). The reviewers also report that ReLU may not be the best activation function choice for DARE. Indeed, it can happen that the neuron outputs before activation are all negative, in this case, increasing the previous weights has the negative-effect of projecting all data (ID and OOD) in the negative reals part, leading to a loss of output diversity after applying ReLU (all outputs are null). However, it seems that, as long as some ID neuron outputs are positive, this negative effect of the ReLU has small impact. We then choose to use ReLU in the experiments as it is the most common activation function. However, as suggested by the reviewers, DARE may benefit from the use of Leaky-ReLU or GELU.

4) We thank the reviewer for reporting the clarity issues in the theoretical part. Their comments will help us improve this section in future work.

After conducting additional experiments in classification, it seems that the DARE approach, in its current form, is not suited for this setup. It would be safer to claim that DARE is a performant method for regression problems with fully-connected networks, which limits the range of application. However, we think that additional experiments in regression are then needed to bring more evidences of the DARE improvements.

We, again, thanks all the reviewers for the time spent reading our paper. Their questions and suggestions will be very helpful for future works.

[1] Zhang, Jingyang, et al. "OpenOOD v1. 5: Enhanced Benchmark for Out-of-Distribution Detection." arXiv preprint arXiv:2306.09301 (2023).

---

### Meta-Review · Area_Chair_TtZY · 2023-11-27

**Metareview:**

The reviewers and meta reviewer all carefully checked and discussed the rebuttal. They thank the author(s) for their honest response.

While the reviewers praised the simplicity and originality of the proposed idea, the submission, as acknowledged by the rebuttal, suffers from several weaknesses and limitations, which all put together warrants further investigations and consolidations.

For example:
* Limited set of experiments (architecture, dataset, scale) and baselines to validate the approach
* Scope of applicability, e.g., with respect to the activation functions, the loss functions and the learning setting (regression vs. classification).
* Clarity around the theoretical analysis

All in all, the paper is recommended for a rejection. We are convinced that the suggestions surfaced through the reviews will help strengthen the paper for a future resubmission, which the reviewers and meta reviewer all encourage.

**Justification For Why Not Higher Score:**

* Rejection unanimously supported by the reviewers
* Unclear scope of applicability
* Missing baselines
* Limited evaluations (in terms of architecture, dataset, scale)

(several of those points were honestly acknowledged by the rebuttal of the author(s))

**Justification For Why Not Lower Score:**

N/A

---

### Decision · Program_Chairs · 2024-01-16

Reject